# Update on Current Microbiological Techniques for Pathogen Identification in Infectious Endophthalmitis

**DOI:** 10.3390/ijms231911883

**Published:** 2022-10-06

**Authors:** Lindsay Y. Chun, Donavon J. Dahmer, Shivam V. Amin, Seenu M. Hariprasad, Dimitra Skondra

**Affiliations:** 1Department of Ophthalmology and Visual Science, The University of Chicago Hospitals and Health System, Chicago, IL 60637, USA; 2College of Medicine, University of South Alabama, Mobile, AL 36688, USA

**Keywords:** ophthalmology, endophthalmitis, retina, microbiology, MALDI-TOF MS, PCR, genome, sequencing, precision medicine, infectious diseases, vision

## Abstract

Infectious endophthalmitis is a vision-threatening medical emergency that requires prompt clinical diagnosis and the initiation of treatment. However, achieving precision in endophthalmitis management remains challenging. In this review, we provide an updated overview of recent studies that are representative of the current trends in clinical microbiological techniques for infectious endophthalmitis.

## 1. Introduction

Infectious endophthalmitis is an ophthalmic emergency that can lead to rapid irreversible vision loss within hours and/or days of clinical diagnosis. The definition of “endophthalmitis” refers to an infection of the intraocular vitreous and/or aqueous humor by a causative agent, such as bacteria or fungi [1]. The vitreous humor is considered the largest component of the eye, and it is composed of a mucilaginous, achromatic, highly hydrated matrix, located in the posterior segment of the orbit between the lens and retina [2]. The aqueous humor is a watery, clear fluid located in the anterior and posterior chambers of the eye, it is secreted from the ciliary body, and exits through the trabecular meshwork located at the iridocorneal angle [3].

Infection of the intraocular compartments can occur in the setting of intraocular surgery, intraocular injections, trauma, contiguous spread from adjacent structures (i.e., keratitis, bleb), and endogenous spread from chronic and transient sources from the bloodstream such as a liver abscess or indwelling central venous catheter, respectively [1]. The progression of endophthalmitis can depend on the inoculum size of the pathogen, virulence factors, and immune activity [1,4,5,6,7,8,9]. A large number of pathogens introduced to the conjunctiva of the eye, such as contaminated surgical solution, can overcome host defenses and increase the risk of endophthalmitis [10]. Membrane-damaging virulence factors such as phospholipases, hemolysins, and sphingomyelinases, can progress invasion leading to increase inflammation, retinal cell damage and ultimately, unsalvageable vision loss [11,12,13]. Therefore, the need to recognize the type of -endophthalmitis with faster and higher yield diagnostic approaches with minute sample sizes is vital to saving vision.

Throughout the past several decades, many different mechanisms of endophthalmitis have been reported including post-cataract, post-vitrectomy, post-keratoplasty, post-injection, posttraumatic, bleb-related, keratitis-related, mold, and endogenous (bacterial, fungal) endophthalmitis [1,14,15]. A major type of endophthalmitis is acute post-cataract endophthalmitis. This form is normally bacterial and presents within a week of cataract surgery. The ocular surface or the flora of the lid skin is routinely the source of infection [1,7]. Yet, occasionally, contaminated surgical instruments or solutions can be the source of outbreaks, as seen in a study reviewing postoperative *Fusarium oxysporum* endophthalmitis after the use of contaminated viscoelastic substances [16].

Intravitreal injection of antibiotics is the current treatment for acute postoperative endophthalmitis, and ceftazidime and vancomycin are commonly used for empirical treatment in presumed bacterial cases [1]. Systemic antibiotics can be used as adjunctive therapy. In 1995, a large prospective trial called the Endophthalmitis Vitrectomy Study (EVS) analyzed patients who received vitrectomy plus intravitreal antibiotics versus only intravitreal antibiotics, in regards to the efficacy of therapy [17,18,19,20]. The study reported that patients who presented with the worst vision had better outcomes when receiving immediate vitrectomy plus antibiotic therapy. In addition, only 20% of the vitrectomy plus intravitreal antibiotics group had residual vision loss compared to 47% of the intravitreal antibiotics only group. A total of 71% of cultures were positive from eyes which had early TAP and antibiotics, whereas only 13% were positive in the vitrectomy plus antibiotics [18].

Chronic post-cataract endophthalmitis is an infrequent type of endophthalmitis mostly caused by fungi or integumentary bacteria, such as *Propionibacterium* acnes. For diagnosis, a negative needle aspirate is commonly seen, thus vitrectomy is needed. For the treatment of chronic P. acnes endophthalmitis, dual therapy including intravitreal antibiotic injections and surgery (removal of intraocular lens, vitrectomy, or capsulectomy) are oftentimes needed due to the high rates (70%) of relapse in chronic cases [19].

Postinjection endophthalmitis is a type of endophthalmitis that occurs after intravitreal injections. Most injections performed today involve the use of an anti-VEGF agent for neovascular age-related macular degeneration, diabetic retinopathy, and many other neovascular proliferative diseases [1]. Each injection causes a 0.05% risk of endophthalmitis that accumulates over the months and years of injections [21,22,23,24,25].

Posttraumatic endophthalmitis occurs in penetrating eye traumas known as open globe injuries [1,24]. The treatment for posttraumatic endophthalmitis includes intravitreal antibiotics (with the addition of topical or systemic antibiotics), vitrectomy, and the removal of any lodged foreign material [1,24].

Vitrectomies are regularly performed for retinal tears, detachments, vitreous hemorrhage, or other retinal conditions [1]. Postvitrectomy endophthalmitis presents less frequently than chronic or acute post-cataract endophthalmitis; however, the pathogens remain similar with the majority of cases being caused by coagulase-negative *Staphylococci,* as well as diagnosis and treatment [1,7]. 

A bleb-related endophthalmitis is an indolent form of endophthalmitis that forms over months to years after glaucoma surgery. The artificially created bleb can become infected leading to endophthalmitis. Blebitis, an infection of bleb, carries a 1% risk over 5 years of progressing to endophthalmitis, according to a study in Japan. Furthermore, leaking blebs increased endophthalmitis by nearly 5-fold [25]. The treatment for bleb-related endophthalmitis is intravitreal and/or intracameral antibiotic injections, in addition to topical antibiotics and systemic antibiotics, such as quinolones, as adjunctive therapy [25]. 

Postkeratoplasty endophthalmitis occurs after a corneal transplant. The combination of intracameral (into aqueous) and/or intravitreal antibiotics (with vitrectomy, if needed) and the replacement of infected cornea is the treatment for post-keratoplasty endophthalmitis [1,26,27,28,29]. 

Lastly, endogenous endophthalmitis is a rare form of endophthalmitis and occurs from bacteremic or fungemic seeding of the eye [1,8]. Unlike the previous types of endophthalmitis, the endogenous form begins in the posterior segment of the eye in the most vascularized layer of the eye, the choroid. Numerous infectious etiologies are associated with endogenous endophthalmitis, such as endocarditis, liver abscess, and urinary tract infections [30,31,32,33,34]. Endogenous bacterial endophthalmitis presents with a complaint of decreased vision, hypopyon, eye pain, and vitritis (cellular infiltration of vitreous) with or without systemic symptoms [33]. Therapy includes intravitreal antibiotics and vitrectomy (if indicated) for endogenous bacterial endophthalmitis and systemic antibiotics for the underlying infection [1,7]. Endogenous fungal endophthalmitis (EFE) is most commonly caused by *Candida albicans* and initially presents as chorioretinitis with “fluffy” white chorioretinal lesions along with overlying vitritis [1,7,34]. A definitive diagnosis is made by vitreous cultures or blood, which may be negative due to transient candidemia. Treatment for *Candida* endophthalmitis and chorioretinitis is determined based on if the chorioretinitis lesions are macula-threatening or non-macula-threatening [1,7,34,35]. Lastly, endogenous mold endophthalmitis (e.g., *Aspergillus* and *Fusarium*) can be seen in immunocompromised and can be treated with intravitreal voriconazole or amphotericin, plus vitrectomy and systemic antifungal therapy [1,35].

However, overall, the routine use of sterile techniques and antimicrobial prophylaxis has rendered infectious endophthalmitis a rare incident, ranging from a rate of 0.012 to 1.3% after cataract surgery, and 0.016 to 0.2% after intravitreal injections [7]. Although endophthalmitis is a rare condition, its incidence is likely to rise with the high frequency of ocular procedures. Cataract surgery and intravitreal injections are among the most commonly performed procedures in ophthalmology, and each procedure involves a risk for infection [36]. Cataract surgery and intravitreal injections are among the most commonly performed medical procedures in the world, and each procedure involves a risk, albeit, a low risk, for infection [9]. As the general population ages, the incidence of cataract surgeries is projected to increase dramatically in developed and developing countries, and the advent of new intravitreal agents for a broad array of retinal diseases, including neovascular age-related macular degeneration and diabetic macular edema, is likely to lead to an increase in intraocular injections performed on a daily basis [37,38,39,40]. Guidelines for preoperative preparation and sterile procedural techniques have likely aided in keeping the reported rates of endophthalmitis low following cataract surgery (0.012 to 1.3%) and intravitreal injections (0.016 to 0.2%) [1,21,22,23,24]. However, the frequency with which these procedures are performed makes the risk of infectious complications a point of serious concern for patient care in ophthalmology. Again, prompt clinical diagnosis and the initiation of treatment are critical to preserving visual function.

Importantly, physicians treating patients with endophthalmitis must be aware of the potentially fatal consequences to vision, and the prognosis for patients can be very poor. In one study, 21.7% of eyes were reported to never being able to regain their baseline visual acuity after 6 months, and in another study up to 10% of eyes were reported to suffer from complete vision loss [1,37,41]. During the time it takes to isolate an organism, the administration of broad-spectrum antibiotics could typically be initiated to salvage the eye. Current recommendations for intravitreal antibiotics include vancomycin (1 mg/0.1 mL) and ceftazidime (2.25 mg/0.1 mL). However, broad-spectrum antibiotics can complicate the course of management because they can put the patient at future risk of succumbing to infection from an antimicrobial-resistant organism.

As hinted throughout, the mantra of “tap and inject” in ophthalmology highlights the keystone components of clinical management in infectious endophthalmitis: the microbiological evaluation of intraocular fluids, and the prompt injection of broad-spectrum intravitreal antimicrobials [1,22,23,24,36]. However, there are real-life caveats to this process. Firstly, nonsurgical biopsies of intraocular fluids can result in inadequate sample volumes or fail, producing a “dry tap”. Secondly, even successfully attained samples may still fail to grow pathogens in a timely fashion, especially if samples are retrieved after antimicrobial administration. These shortcomings are further compounded by the insufficient and suboptimal results attained from standard microbiological methods that are currently the “gold standard”. Gram stains of intraocular samples are negative in 50–60% of cases, and negative cultures rates are up to 60% from aqueous tap biopsies, and up to 55% in vitreous tap biopsies [1,42,43,44]. The reported rates for positive cultures that result from samples obtained from surgical vitrectomy range widely, from 44.6 to 90% of cases [1]. Although it is routine to administer broad-spectrum intravitreal antibiotics before pathogen identification, the impending emergence of antimicrobial resistance poses a serious epidemiological concern [9,45,46,47]. Thus, accurate and rapid pathogen identification and their antimicrobial susceptibility profiles is a clinically relevant and important area of investigation.

The challenge to yield timely, precise, and accurate identification of diverse pathogens, and establish antimicrobial susceptibility profiles from small intraocular sample volumes, even after antimicrobial administration, guides active areas of clinical research in accordance with the general trends toward precision medicine. Herein, we will review recent studies representing a portion of the expansive literature on infectious endophthalmitis to provide an updated overview of microbiological techniques that have been developed for pathogen identification in infectious endophthalmitis (Table 1).

## 2. Nucleotide Based Methods: Can PCR Prevail When Cultures Fail?

The clinical trends towards “precision medicine”, especially within infectious diseases, utilizes the massive advancements made in genomic technology to rapidly characterize an infection and guide treatment and prevention. The below studies investigate the value of genome-centralized PCR (polymerase chain reaction) methodologies for pathogen identification in infectious endophthalmitis.

### 2.1. Specific PCR and Quantitative PCR

Kosacki et al. published a large, prospective study based in France, that compared the utility and rates of successful pathogen identification by: (1) standard culture, (2) 16S rRNA panbacterial PCR, and (3) quantitative PCR (qPCR, also called real time or rtPCR) [7]. For 16S rRNA panbacterial PCR, samples were amplified and sequenced using the “universal” 16S rRNA primer sequence common to all known bacteria. For qPCR, also called real time or rtPCR, specific custom primers were used to perform a targeted amplification and quantification of bacterial pathogens.

This study included 284 intraocular samples (aqueous taps of 150–200 μL, vitreous taps of 200–300 μL, and vitrectomy biopsies of 500 μL) from 153 pts with delayed onset postoperative endophthalmitis. For analysis with standard microbiological culture, samples were inoculated in pediatric blood culture bottles for 14 days and subsequently plated for phenotypic identification. For PCR analysis, samples with enough residual volume after culture were processed for DNA extraction.

From samples analyzed prior to intravitreal antibiotics, the rates of successful pathogen identification were: 77/142 (54%) with standard culture, 67/137 (49%) with 16S rRNA PCR, and 8/120 (7%) with qPCR. 6/25 A total of 24% of culture-negative cases had positive 16S rRNA results. From samples analyzed after intravitreal antibiotics (vancomycin and ceftazidime), the rates of successful pathogen identification were: 45/124 (36%) with standard culture and 57/120 (48%) with 16S rRNA PCR.

The qPCR tests demonstrated that the bacterial load of samples before and after intravitreal antibiotics did not significantly differ. The 16S rRNA panbacterial PCR was found to have lower sensitivity and specificity than qPCR tests and a longer turnaround time.

The turnaround time for results via 16S rRNA panbacterial PCR was 2–3 days, and that for qPCR was 2–3 h. Standard microbiological culture was greater than 14 days due to the initial step of growth in blood culture bottles.

Additionally, this study showed no significant association between the microbiological profile of the samples and vision prognosis; however, higher bacterial load in vitreous humor samples was associated with worse vision prognosis.

The findings from this study by Kosacki et al. suggest that in the event of negative culture results, 16S rRNA PCR may allow for positive pathogen identification. Although the majority of the identified pathogens were *S. epidermidis* (65% of cases), the study does not report the concordance of pathogen identification between the different methodologies. This study also touches upon a major limitation in microbiological studies of intraocular fluids: small sample volume. The strategy of incubating intraocular samples in blood culture bottles may be worth refining for small endophthalmitis samples, allowing for a higher yield of pathogen in more abundant amounts for subsequent analysis with nucleotide-based or phenotype-based studies for identification. This larger sample yield may allow for antibiotic susceptibility profiles as well from the same sample source.

### 2.2. Specific Targeting of Commonly Implicated Pathogens Pathogens: Multi-Mono PCR (mmPCR) and Target High-Throughput Sequencing

Interestingly, van Halsema et al. studied the ability of a technique utilizing a more targeting PCR approach called “multi mono PCR” (mmPCR), in which primers against 20 common bacterial pathogens causing infectious endophthalmitis were used for pathogen identification [8]. In this study, mmPCR was applied with 20 rtPCRs simultaneously run, using a set of target genes of 20 suspected bacterial pathogens, and a “panbacterial” target for the 16S rRNA gene common to all bacterial species not among the 20 catalogued pathogens. However, all the samples in this study were obtained from patients prior to intravitreal administration of spectrum antibiotics.

In total, 27 samples of vitreous biopsies prior to antibiotics containing 200 μL fluid were stored at −80 °C, mixed with lysis buffer, and sent to a lab for DNA extraction and analyzed with mmPCR and standard culture. The mmPCR had 24/27 (89%) concordance with standard culture in species-level identification. The mmPCR had a similar sensitivity and specificity profile as culture, if not slightly lower (sens 91%, spec 94%). The authors suggest that this strategy of mmPCR may potentially help identify patterns for antimicrobial resistance in the future, but it remains unclear how antibiotics prior to biopsy acquisition may have affected the pathogen profiles obtained.

In a similar, but larger-scale application of species-specific PCR, Gandhi et al. studied the use of targeted high-throughput sequencing (HTS, synonymous with “next gen sequencing”) for pathogen identification in infectious endophthalmitis [9]. In their methodology, the authors selectively amplified pathogenic genomic regions with the use of panels of known genes, and subsequent massive parallel sequencing-a process they refer to as targeted HTS—was used as an initial step. The authors stated that this stepwise approach gave the advantage of avoiding amplification of potential host nucleic acids that could confound pathogen identification. The speed of sequencing can take from 2–8 h, depending on the materials used [36,48].

In total, 75 patients with infectious endophthalmitis had 0.5 mL undiluted vitreous tap followed by intravitreal antibiotics and amphotericin B. Samples were analyzed with standard microbiological culture and targeted HTS. DNA was extracted from these vitreous tap biopsies with commercial kits, and PCR was performed using a primer for the 16S rRNA panbacterial target, and a primer for the fungal small subunit rRNA target ITS2.

In total, 18/75 samples were culture-positive-15/18 (83.3%) grew bacteria, and 3/18 (16.7%) grew fungi that were identified at the species level. Targeted HTS had species identification concordance with culture in 14/15 (93.3%) bacterial and 3/3 (100%) fungal cases. From 57/75 samples that were culture-negative, targeted HTS identified fungal pathogens in 36/57 (63.1%) and bacterial pathogens in 11/57 (19.3%), including 5/57 (8.8%) combined fungal and bacterial. However, microbes were detected in 4/70 (5.7%) noninfectious control samples via targeted HTS; this could represent either contamination, false positive identification, or a subclinical disease-related biomarker. Due to sample volume limitation, these samples were not tested with other PCR-based tests for further confirmation. HTS provided positive pathogen identification from intraocular samples that had negative culture results.

In a similar fashion, Mishra et al. took 16 culture-negative vitreous samples, purified DNA using commercial kits, and targeted seven regions of the 16S rRNA gene for amplification and metagenomic analysis against the genomic bank database [42]. The HTS/NGS method was able to identify pathogens for every sample of culture-negative endophthalmitis. They were able to identify multiple pathogenic species in all samples, ranking each species by the predominance of its genetic material in each sample, suggesting that this method could potentially identify polymicrobial infections and/or fastidious organisms that cannot be readily identified via standard culture methods.

This study demonstrates that HTS may be helpful in more quickly identifying fastidious organisms, such as fungal pathogens and potentially polymicrobial infections. This is especially relevant in regions where endophthalmitis caused by fungi or atypical pathogens may be more common [43]. However, the multiple processing steps in HTS to isolate and amplify DNA targets from samples increase the likelihood of contamination, and necessitates a predetermined threshold by which signal vs. noise parameters are set. The large amount of sequencing involved would also be costly and require a large amount of labor. Additionally, these studies did not report the timespan between sample acquisition and result; the reliance on an external specialized lab to perform this highly specialized analysis would logistically prevent rapid pathogen identification.

### 2.3. Whole Genome Sequencing: Identify the Pathogen without a Priori Determined Target Gene

Lee et al. have described the use of whole genome sequencing (WGS) for direct pathogen identification and quantification in endophthalmitis, while bypassing the use of genetic target panels [44,45]. The authors suggest that by eliminating the reliance on predetermined target genes for amplification and sequencing, there may be less bias towards the identification of a priori suspected pathogens, and the potential for finding either novel or unusual pathogens involved in the disease process. The authors state that 1 nanogram of DNA is sufficient for WGS analysis, and time to result is approximately 24 h after the DNA sample has been processed.

In this study, 50 intraocular fluid samples from cases of post-procedural endophthalmitis were submitted for standard culture and for WGS. For WGS, samples were stored at −80 °C prior to DNA extraction using commercial kits, and yielded 2–8 μL of DNA per sample. Each sample was enzymatically processed into smaller DNA base-pair fragments which are then put through massive parallel sequencing and simultaneous matching against genes of DNA-based organisms registered in the NCBI Genbank. This automated process also allows for the determination of bacterial load in each sample, based upon the number of times a genetic sequence appeared. Of the 24 culture-positive cases, WGS had concordance of 17/20 (85%) with culture results; of the 26 culture-negative cases, WGS had an 8/22 (36%) rate of potential pathogen identification. The authors also found associations between worse visual outcomes in patients and higher bacterial load or identification of pathogens other than S. epidermidis.

Additionally, the presence of the Torque teno virus and Merkel cell polyomavirus was detected in 49% and 19% of samples, respectively, a finding that the authors suggested may provide future steps for biomarker identification in endophthalmitis, but the significance of which has not yet been elucidated [1].

WGS appears to particularly demonstrate promise for the identification of pathogens in culture-negative cases, likely where there are fastidious organisms or not enough live and reproducible pathogenic material to grow on culture. In a separate but similar pilot study by the same group, 16S rRNA PCR also showed no pathogen identification in culture-negative cases, where WGS was able to identify Torque teno virus, suggesting the more powerful ability of WGS to attain genetic matches from otherwise pathogenically empty samples.

However, the relatively low concordance between standard culture results and WGS indicates the potential current lack of accuracy in this technology. It is possible that WGS amplifies and identifies background contaminants incorrectly as the main pathogens, and may require improved thresholds to differentiate probable versus background identification. Whether the samples in the study were from patients who had already received broad spectrum antibiotics, thus altering the contents of analyte, is also unclear. The ability of WGS to determine antibiotic susceptibility patterns and minimum inhibitory concentrations remains unexplored. Other limitations of WGS, including its ability to identify solely DNA-based organisms and its core reliance on the Genbank database, and improved identification rates with higher abundance of DNA in any given sample, are shortcomings that are similar to those of most other pathogen-identifying methods. In vitro experiments may allow for clarity of the limits and abilities of WGS.

## 3. MALDI-TOF MS: Using Proteomic Fingerprints to Find the Culprit Pathogen

Matrix-Assisted Laser Desorption/Ionization Time of Flight Mass Spectrometry (MALDI-TOF MS) is a method by which a proteomic profile of a pathogen is created with the mass-to-charge qualities of its peptide components, and matches the sample against a proteomic “fingerprint” database of a wide array of organisms [46,47]. The MALDI-TOF MS laser ionizes whole cell extracts from colonies grown in culture to produce a peptide fingerprint profile, and compares the profile against a proteomic database to identify a pathogen to a species level [50,51,52]. Similar to nucleotide-based methods, this method of proteomic “fingerprint-matching” can allow for the identification of gram-negative and gram-positive bacteria, aerobes, anaerobes, mycobacteria, *Nocardia* spp., yeasts, filamentous fungi, and viruses. Furthermore, if there is adequate sample volume, the turnaround time from sample acquisition to results can be a matter of minutes to hours, compared to the days required for cultures.

Currently in the US, MALDI-TOF MS is used in the clinical laboratory setting as a confirmatory test to identify organisms after they have already grown from culture. Previous studies have looked into the use of MALDI-TOF MS on infected human samples; urine and spinal fluid samples have previously been directly analyzed with MALDI-TOF MS for pathogen identification from cerebrospinal fluid (CSF) and urine from cases of meningitis and urinary tract infections, respectively, without prior culture [41,51]. As MALDI-TOF MS requires between 10^3^ and 10^4^ cells per analyte for positive identification [46], the relatively larger yield obtainable from urine and CSF samples are particularly appropriate for direct analysis. Compared to traditional identification methods, MALDI-TOF MS has been shown to have high potential as an analytical tool for the characterization of different types of microorganisms, and has a gain of time in days [37,47,50].

There have been several published studies on the comparative results of pathogen identification between conventional microbiological techniques to MALDI-TOF MS, from patient samples of clinically suspected endophthalmitis. Pathogen identification rates have been reported to be between 65.9% [39]. Angrup et al. utilized MALDI-TOF MS to identify *Stenotrophomnas maltophilia* as the causative pathogen from vitrectomy samples in a case of endophthalmitis outbreak from a contaminated vial of bevacizumab that was used for intravitreal injections in northern India [53]. The successful confirmatory identification of the pathogen using MALDI-TOF MS allowed the physicians to change the treatment of the patients from IV ciprofloxacin to IV and intravitreal ceftazidime, underscoring the importance of accurate pathogen identification to improve patient outcomes. However, most of these studies analyzed endophthalmitis samples with MALDI-TOF MS after first growing and amplifying pathogens for up to 24 h via inoculation of pediatric blood culture bottles, or other broth medium with infected vitreous humor [37,38,39,40,53,54]. The potential for the direct analysis of intraocular samples with MALDI-TOF MS would bypass the extra processing time and resources of growing organisms in blood culture bottles.

The mammalian eye has the feature of acting as its own culture bottle with, albeit limited, nutritional media, and can serve as a reservoir for pathogenic growth. Stemming from this principle, the potential utility of MALDI-TOF MS for direct analysis of infectious endophthalmitis samples, was previously explored in a proof-of-concept study by the authors of this review [49]. In an in vitro model of bacterial endophthalmitis, vitreous humor aspirated from freshly enucleated porcine eyes was inoculated and incubated at 37 °C with different inocula of *S. aureus*, and minimally processed with centrifugation to form bacterial pellets for direct analysis with matrix solution α-cyano-4-hydroxycinnamic acid. MALDI-TOF MS achieved accurate pathogen identification from direct analysis of intraocular samples with confidence values of up to 99.9%. Time from sample processing to pathogen identification was <30 min. The minimum number of bacteria needed for positive identification was 7.889 × 10^3^ colony forming units (cfu/μL). Of course, this in vitro model has limited clinical implications; using only one strain of a gram-positive bacterium limits the demonstrable applicability of MALDI-TOF MS in cases of endophthalmitis caused by other microbial pathogens. As this was an in vitro experiment, the in vivo conditions of endophthalmitis are not fully replicated. The in vitro replication of endophthalmitis will serve only as a limited model of the pathogenesis of endophthalmitis.

The ability of MALDI-TOF MS to identify pathogens after antimicrobial administration was also explored in a case report of bleb-related endophthalmitis, in which the patient had already been administered intravitreal broad-spectrum antibiotics prior to pars plana vitrectomy [55]. Briefly, vitreous samples from pars plana vitrectomy were sent for standard Gram stain and culture, and the remaining samples were prepared for MALDI-TOF MS analysis by centrifuging them at 6000× *g* for 10 min at 4 °C to produce concentrated visible bacterial pellets. The bacterial pellets were washed with sterile double-distilled H2O. The pellets appeared opaque and white, with mucoid consistency. Sterile plastic loops were used to apply the bacterial pellets on the spots of the target plate for MALDI-TOF MS. A colony of E. coli was used as a positive control. Each spot was overlaid with a matrix of α-cyano-4-hydroxycinnamic acid per protocol (Vitek MS, bioMérieux), and the target plate was inserted into the machine. MALDI-TOF MS gave rapid identification of an organism, *Gemella sanguinis,* with 99.7% confidence value, while conventional cultures gave no results. *G. sanguinis* is a Gram-positive anaerobe first characterized in 1998. It has shown sensitivity to vancomycin and cephalosporins, which were used empirically in our case [56,57]. *G. sanguinis* is an anaerobic species that was relatively recently discovered, possibly indicating it may be a fastidious organism to grow via conventional methods [58]. Though rare, cases of endophthalmitis by *Gemella* species have previously been reported, typically in the setting of immunocompromised status [59]. Cases of BRE in the US are more commonly caused by streptococci, enterococci, and *Haemophilus influenzae*; however, this would be the first reported account of bleb-related endophthalmitis by *G. sanguinis* [1,52]. In this case report, the conventional techniques of Gram stain and culture grew no organisms likely due to small sample volume and low bacterial load acquired from vitreous and aqueous aspirates, as well as the prior administration of antibiotics. Future experimentation using in vivo models of endophthalmitis and modifications of sample preparation volumes used, dilution process, centrifuging parameters, and concentrations of MALDI reagents used for bacterial extraction, would be needed to develop standardized protocol for direct analysis of intraocular samples.

MALDI-TOF MS can also provide antimicrobial susceptibility testing (AST) and values for minimum inhibitory concentration (MIC) by analyzing the molecular products (ranging from peptides to glycolipids) from pathogens, after they are exposed to different antimicrobial agents. The time-to-result from sample harvest of *B. fragilis* grown from blood culture bottles to MIC result was 3 h in one in vitro study [60]. However, this biomarker identification method would be limited to the specific resistance patterns included in the database [61].

A separate commercially available antimicrobial susceptibility test (AST) system called VITEK 2 utilizes fluorescence-based technology to analyze the susceptibility profiles of Gram-positive and Gram-negative bacteria [62,63]. VITEK 2 is clinically used to analyze microorganisms that are grown in cultures and have already been identified from culture, per standard techniques. VITEK 2 AST has been shown to have a high degree of agreement with standard methods for determining the minimum inhibitory concentration (MIC) of antibiotics, with a gain-of-time of hours to days, and high reproducibility [62,63]. The authors of this review demonstrated that VITEK 2 can be used to directly analyze vitreous humor samples from in vitro models of *S. aureus* endophthalmitis without prior culture, with up to 94.4% concordance with conventional methods [49]. Time to result attainment was 8–9.25 h with direct analysis with VITEK 2, in contrast to the multiple days needed via conventional methods [37,38]. Our findings showed that the growth of the causative pathogen through standard culturing methods is not necessary for analysis with VITEK 2, given that the minimum turbidity (0.5 McFarland units) of the analyzed material is met.

Similar to other methodologies discussed, the scope of pathogen identification by MALDI-TOF MS is limited by the breadth of organisms established in the database of the specific biotyper software that is employed, and the yield of pathogens in an analyte. MALDI-TOF MS is also not suited for the analysis of polymicrobial infections, although polymicrobial endophthalmitis is incredibly rare and this limitation may have low relevance. The accessibility of the machinery and software would also be a major roadblock for certain communities and health systems. A wider breadth of organisms including other bacteria, mycobacteria, fungi, and polymicrobial infections, should also be investigated with MALDI-TOF MS. Well-designed in vivo animal models and analysis of human samples of endophthalmitis are needed to validate the clinical value of MALDI-TOF MS and automated AST for direct analysis without prior culture. The scope of pathogen identification is limited by the breadth of organisms established in the database of the specific biotyper software that is employed. Additionally, for clinical applicability of the techniques we describe, there must be an adequate quantity of bacteria present in intraocular samples obtained from patients. Further, the exact effect of antimicrobials in endophthalmitis samples for analysis with MALDI-TOF MS, for example, the potential changes to the microbial epitopes and molecular byproducts that may influence the peptide profile upon analysis, has yet to be fully explored; unlike conventional microbiological methods, which require that the organism be intact or alive for proper identification, MALDI-TOF MS only requires the presence of particles of the culprit organism.

Nonetheless, rapid, and accurate diagnostic approaches for endophthalmitis are crucial, and current diagnostic methods have numerous limitations. Although further studies and optimization models for the direct analysis of human ocular samples from cases of endophthalmitis with MALDI-TOF MS are needed, the ability to identify microorganisms without prior culture could represent a novel and innovative shift in clinical microbiological methodology. The ability of MALDI-TOF MS to identify fastidious and relatively rare organisms also demonstrates that there are perhaps ways to improve the gold standard of pathogen identification to become more rapid, accurate, and use fewer resources. This is certainly of high clinical utility and significance in an era when the value of targeted molecular therapies would help to safeguard against increasing antibiotic resistance, and its consequent morbidity and costs to healthcare [1].

## 4. Conclusions

Pathogen identification in infectious endophthalmitis poses big challenges in clinical microbiology. The intraocular compartments too often provide inadequate biopsy volumes, and the standard administration of intravitreal antibiotics can prevent the subsequent isolation of live organisms. Importantly, the rapid progression of disease and threat to vision makes timely pathogen identification vital for optimal care. The studies described above, exploring PCR, HTS, whole genome sequencing, and MALDI-TOF MS, cover a portion of the microbiological advancements made within the past several years. All strategies hold their merits and shortcomings in the common goal to maximize clinical information with the minimum materials necessary. Perhaps it may one day be within reach for us to utilize the strongest components of these novel molecular techniques to help develop an optimized method that can rapidly provide accurate identification of a wide array of pathogens from small sample volumes, and withstand the effects of antimicrobials or immune-mediated modifications, while requiring minimal processing protocols and accessible analytical technology.

## Figures and Tables

**Table 1 ijms-23-11883-t001:** Summary of microbiological techniques for pathogen identification in endophthalmitis.

Technique	Volume of Analyte Required	Rate of Pathogen Identification with Technique Described	Approximate Time-to-Result from Sample Acquisition to Pathogen Identification	Rate of Pathogen Identification with Standard Gram Stain and Culture	Reference
16S rRNA panbacterial PCR	150–500 μL for DNA extraction	67/137 (49%) bacteria in culture-positive samples	2–3 days	77/142 (54%) bacteria	Kosacki et al., 2020 [7]
	150–500 μL for DNA extraction	6/25 (24%) bacteria in culture-negative samples	2–3 days	0/25 (0%)	
quantitative PCR (qPCR or rtPCR)	150–500 μL for DNA extraction	8/120 (7%) bacteria from all samples	2–3 h	77/142 (54%) bacteria	
Multi-mono PCR (mmpCR)	200 μL for DNA extraction	24/27 (89%) bacteria in culture-positive samples	90 min (time for rtPCR)	27/27 (100%) bacteria	van Halsema et al., 2021 [8]
Targeted High-throughput sequencing	500 μL for DNA extraction	14/15 (93.3%) bacteria, 3/3 (100%) fungi in culture-positive samples	2–8 h	15/18 (83.3%) bacteria, 3/18 (16.7%) fungi	Gandhi et al., 2019 [9], Reuter et al., 2015 [36], Mellman et al., 2011 [48]
	500 μL for DNA extraction	11/57 (19.3%) bacteria, 36/57 (63.1%) fungi in culture-negative samples	2–8 h	0/57 (0%)	
Whole genome sequencing	1 ng DNA	17/20 (85%) bacteria in culture-positive samples	24 h	24/24 (100%) bacteria	Lee et al., 2020 [44]
	1 ng DNA	8/22 (36%) bacteria in culture-negative samples	24 h	0/22 (0%)	
MALDI-TOF MS in vitro direct analysis	10^3^ to 10^4^ cells per analyte	12/14 (85.7%) bacteria in samples with visible pellet with 96.1–99.9% confidence value	30 min	N/A; 2/2 (100%) bacterial control	Chun et al., 2019 [49]
MALDI-TOF MS analysis of culture-grown samples		65/90 (72%) to 37/41 (90.2%) bacteria in culture-positive samples	5 days for culture growth + 14.4 h protein extraction for MALDI-TOF MS	41/41 (100%) bacteria	Tanaka et al., 2017 [37]
		29/44 (65.9%) bacteria in all samples	3.17 days	20/44 (45.5%) bacteria	Xu et al., 2020 [39]

## Data Availability

Not applicable.

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
