# Peer review of "Update on Current Microbiological Techniques for Pathogen Identification in Infectious Endophthalmitis"

_ijms, 2022, doi:10.3390/ijms231911883_

Round 1

Reviewer 1 Report

Congratulations for the clear and concise review on current microbiological techniques for identification of pathogen agent in infectious endophtalmitis 

1. (line 114-115, page 3): I suggest only rephrase the sentence  to make it clearer : "71% of cultures were positive  in the antibiotics only, whereas only 13% were in the vitrectomy plus antibiotics [18]". 71% of cultures were positive from eyes which had early TAP and antibiotics, whereas only 13% were positive in in the vitrectomy plus antibiotics

2. Page 4,line 195 : endogenous endophtalmitis  intead of endophtalmites

Author Response

Thank you for your review and comments. We have incorporated your suggestions in the new edited manuscript.

Reviewer 2 Report

Review article on laboratory  methods to diagnose infectious bacterial  endophthalmitis and how they compare with standard culture techniques. 

Worthy topic, but  review not well organized or focused.  Would suggest less information on the clinical picture ( introduction) , other than pointing out the need for faster and higher yield diagnostic approaches with small sample sizes available. 

Laboratory methods:  This section should be key of this review.  Suggest better descriptions of each lab technique and a table of pro and cons for each.  Tables showing results from published series may best be compared if presented in tabular form  with text material added only if needed. 

Key issues for each technique should be clearly listed ( maybe table) and in summary list the key issues to be addressed in evaluating these and any future methods for laboratory diagnosis of intraocular eye infections. 

In summary , good topic for a review article, but suggest better focus, adding tables to improved clarification of key points to be made. 

Author Response

Thank you for your comments and review. We have modified the table and reorganized the introductory material per your recommendations.